# Proton Beam Therapy versus Photon Radiotherapy for Stage I Non-Small Cell Lung Cancer

**DOI:** 10.3390/cancers14153627

**Published:** 2022-07-26

**Authors:** Yang-Gun Suh, Jae Myoung Noh, Doo Yeul Lee, Tae Hyun Kim, Unurjargal Bayasgalan, Hongryull Pyo, Sung Ho Moon

**Affiliations:** 1Proton Therapy Center, Research Institute and Hospital, National Cancer Center, Goyang 10408, Korea; suhmd@ncc.re.kr (Y.-G.S.); ldy8816@ncc.re.kr (D.Y.L.); k2onco@ncc.re.kr (T.H.K.); unurjargal@cancer-center.gov.mn (U.B.); 2Department of Radiation Oncology, Samsung Medical Center, Sungkyunkwan University School of Medicine, Seoul 06351, Korea; rodrno@skku.edu; 3Department of Radiation Oncology, National Cancer Center, Ulaanbaatar 13370, Mongolia

**Keywords:** proton beam therapy, stereotactic ablative radiotherapy, stereotactic body radiotherapy, hypofractionated radiotherapy, non-small cell lung cancer, dosimetric comparison

## Abstract

**Simple Summary:**

Stereotactic body radiotherapy (SABR) is accepted as a standard of care for patients who are not candidates for surgery in stage I non-small cell lung cancer (NSCLC). SABR has shown encouraging disease control and acceptable toxicity in peripherally located stage I NSCLC. However, for centrally located tumors around the proximal bronchial tree or for tumors located close to the chest wall, toxicities by SABR are not negligible. Therefore, proton beam therapy (PBT), which provides better organ at risk (OAR) sparing than photon radiotherapy by the Bragg peak, was tested and investigated to reduce radiation-induced toxicities in stage I NSCLC. Here, we compared 112 and 117 stage I NSCLC patients who underwent PBT and photon radiotherapy, respectively. PBT showed significantly lower lung and heart radiation exposure than photon radiotherapy without worsening disease control. PBT could be an effective treatment to reduce long-term toxicities of the lung and heart.

**Abstract:**

Proton beam therapy (PBT) and photon radiotherapy for stage I non-small cell lung cancer (NSCLC) were compared in terms of clinical outcomes and dosimetry. Data were obtained from patients who underwent PBT or photon radiotherapy at two institutions—the only two facilities where PBT is available in the Republic of Korea. Multivariate Cox proportional hazards models and propensity score-matched analyses were used to compare local progression-free survival (PFS) and overall survival (OS). Survival and radiation exposure to the lungs were compared in the matched population. Of 289 patients included in the analyses, 112 and 177 underwent PBT and photon radiotherapy, respectively. With a median follow-up duration of 27 months, the 2-year local PFS and OS rates were 94.0% and 83.0%, respectively. In the multivariate analysis, a biologically effective dose (BED_10_, using α/β = 10 Gy) of ≥125 cobalt gray equivalents was significantly associated with improved local PFS and OS. In the matched analyses, the local PFS and OS did not differ between groups. However, PBT showed significantly lower lung and heart radiation exposure in the mean dose, V5, and V10 than photon radiotherapy. PBT significantly reduced radiation exposure to the heart and lungs without worsening disease control in stage I NSCLC patients.

## 1. Introduction

Stereotactic ablative radiotherapy (SABR) has shown promising disease control and acceptable toxicity for stage I non-small cell lung cancer (NSCLC) [1,2,3]. However, regarding cases of centrally located tumors around the proximal bronchial tree, several prospective studies have demonstrated significant toxicities such as bronchopulmonary hemorrhage, fistula, or pneumonitis [4,5]. As a result, radiotherapy regimens longer than 3–5 fractions, which are usually used for SABR, are also used for centrally located stage I NSCLC [5,6,7]. Furthermore, SABR for tumors located close to the chest wall can result in chest wall toxicities, including chest wall pain and rib fracture, which, though not life-threatening, deteriorate patients’ quality of life [8,9].

The proton beam has a unique feature—the Bragg peak, which is a plot of the rapid energy loss of fast protons in the material of the beam path. Therefore, proton beam therapy (PBT) provides better organ at risk (OAR) sparing than photon radiotherapy, by preventing unnecessary radiation. Several studies reported durable clinical outcomes and the dosimetric superiority of PBT in stage I NSCLC [10,11,12,13]; however, several issues, including range uncertainties, interplay effects, and uncertainties in biological effectiveness, potentially affect its efficacy and safety in lung cancer compared to photon radiation [14,15]. In addition, the Bragg peak is usually degraded in lung tissue because of the extremely low density of the lung tissue, which is attributable to its air-filled spaces [16]. These issues are usually considered negligible or controlled by robust planning for PBT, especially in passive scattering proton beam therapy (PSPT) [16,17]. However, the clinical significance of these issues remains uncertain, as clinical outcomes of PBT have not been compared with those of photon radiation therapy in prospective randomized clinical trials.

In this study, we hypothesized that PBT may cause reduced treatment-related toxicities and improve dosimetric outcomes compared to photon radiotherapy for stage I NSCLC without deteriorating disease control. To compare between PBT and photon radiotherapy, we evaluated outcomes of patients with stage I NSCLC treated with PBT and photon radiotherapy at the only two institutions where PBT is available in the Republic of Korea.

## 2. Patients and Methods

### 2.1. Patients

Medical records were reviewed for patients with stage I NSCLC (tumor size ≤4 cm, N0, American Joint Committee on Cancer Staging, 8th edition) treated with definitive radiotherapy at the National Cancer Center (NCC) and Samsung Medical Center (SMC) in the Republic of Korea between February 2015 and June 2019. This multi-institutional retrospective study was approved by the institutional review boards of both institutions. The requirement for written informed consent was waived owing to the retrospective nature of the study. All patients were medically inoperable or refused to undergo surgery. The staging workup included chest computed tomography (CT), magnetic resonance imaging of the brain, positron emission tomography (PET)/CT, and pulmonary function tests according to the appropriate indicators of lung cancer management by the Health Insurance Review and Assessment Service (HIRA) of the Republic of Korea [18]. For patients whose pathological confirmation of diagnosis was unavailable due to tumor location or prohibitive risk of the percutaneous needle biopsy, the diagnosis of lung cancer was made by a multidisciplinary tumor board at each institution, considering imaging studies and patients’ clinical histories. Simulation, target volume definition, treatment planning, and delivery of PBT were performed at the NCC [12] and SMC [19], as previously described.

### 2.2. Statistical Analysis

Local failure was defined as regrowth (an increase in diameter by at least 20%) of the target lesion on chest CT, accompanied by increased fluorodeoxyglucose (FDG) uptake on PET/CT. Adverse events were graded according to the National Cancer Institute-Common Terminology Criteria for Adverse Events, version 5.0. The Shapiro–Wilk test was used to determine the normality of the distribution for continuous variables. The Mann–Whitney U test or *t*-test for continuous variables and the Pearson chi-square test or Fisher’s exact test for categorical variables were used to analyze differences in clinical features and treatment variables between patients treated with PBT and photon radiotherapy. Survival times were calculated as the interval between the first day of radiotherapy and the occurrence of the first event, be it death or disease recurrence. Survival was evaluated using Kaplan–Meier estimates, and comparisons were conducted using the log-rank test. To compare multiple groups, *p*-values were adjusted using the Benjamini–Hochberg method. Multivariate Cox proportional hazards regression with backward elimination was used to identify prognostic factors affecting local progression-free survival and overall survival. The dose–response analysis for local control was performed with the generalized logistic regression based on categories of BED_10_. A *p*-value less than 0.05 was considered statistically significant (*p* < 0.05).

### 2.3. Propensity Score-Matched Analysis

Propensity score-matched analysis was performed to adjust for potential bias associated with prognostic factors related to treatment (PBT versus photon radiotherapy). The propensity score was calculated using a logistic regression model that included the following covariates: age, sex, European Cooperative Oncology Group (ECOG) performance status (0, 1 versus 2, 3), Charlson comorbidity index (0, 1 versus ≥2), T stage (T1 versus T2a), pathology (adenocarcinoma versus others), biological effective dose (BED_10_) (<125 Gy versus ≥125 Gy), and tumor location (central versus peripheral). The propensity score was used to match each patient treated with PBT to one patient treated with photon radiotherapy, using nearest neighbor matching without replacement, and a caliper width that is 0.25 times of the standard deviation (SD). The absolute standardized mean differences were used to assess the balance of covariates between the two groups in the matched dataset. All statistical analyses were conducted using the R statistical software (version 4.1.3; R Foundation for Statistical Computing, Vienna, Austria). Propensity score matching and balance tests were performed using the R MatchIt and Cobalt packages, respectively.

## 3. Results

### 3.1. Patients and Treatment

A total of 289 patients (93 with NCC and 196 with SMC) were included in the analysis. Among them, 112 (49 with NCC and 63 with SMC) and 177 (44 with NCC and 133 with SMC) patients were treated with PBT and photon radiotherapy, respectively. The median age of all patients was 76 years (interquartile range (IQR), 72–80); males numbered 230 (79.6%). The Eastern Cooperative Oncology Group (ECOG) performance status score of 2–3 was significantly higher in patients treated with photon radiotherapy compared to those treated with PBT. The characteristics of all patients and subgroups according to treatment are shown in Table 1.

The most common histological finding was squamous cell carcinoma (*n* = 89, 30.8%), followed by adenocarcinoma (*n* = 88, 30.4%). A total of 76 patients (26.3%) had centrally located tumors, whereas 213 patients (73.7%) had peripherally located tumors, including those close to the chest wall (*n* = 157, 54.3%). T2a tumors were observed in 83 patients (28.7%).

The median total radiation dose was 60 (range, 48–70) Gy, fraction 4 (range, 4–22), and a biologically effective dose (BED_10_, using α/β = 10 Gy) of 150 cobalt gray equivalents (CGE) (range, 78–150). Most patients (*n* = 160, 55.4%) received a BED_10_ of >125 CGE. Of the 76 patients with centrally located tumors, a radiation dose of 60 CGE in 15 fractions (BED_10_, 84 CGE) was most commonly prescribed (*n* = 15, 19.7%), and 60 CGE in 20 fractions (BED_10_, 78 CGE), 64 CGE in 8 fractions (BED_10_, 115.2 CGE), and 60 CGE in 4 fractions (BED_10_, 150 CGE) were prescribed in 13 (17.1%), 9 (11.8%), and 9 (11.8%) patients, respectively. For patients with tumors close to the chest wall, 60 CGE in 4 fractions, 50 CGE in 4 fractions (BED_10_, 112.5 CGE), and 64 CGE in 8 fractions were predominantly prescribed in 94 (59.9%), 20 (12.7%), and 13 (8.3%) patients, respectively. In 56 patients with other peripherally located tumors, almost all patients (*n* = 50, 89.3%) received a radiation dose of 60 CGE in four fractions. The details of radiation dose prescription according to tumor location and radiation modality are shown in Appendix A. The rate of patients receiving a BED_10_ of >125 CGE was higher in patients treated with photon radiotherapy compared to those treated with PBT (66.1% versus 38.4%, *p* < 0.001). The tumor characteristics and prescribed radiation doses are summarized in Table 2.

### 3.2. Survival

The median follow-up duration for all patients was 27 months (IQR, 17.7–39); it was 27 (IQR, 18–38) and 27 (IQR, 17–40) months for patients with photon radiotherapy and PBT, respectively. Eighty-three patients developed disease progression. Among these patients, local progression, regional failure, and distant metastasis were observed in 18, 28, and 57 patients, respectively. Among 28 regional failures, 5 occurred in the same lung lobe and 24 occurred in regional lymph nodes. The contralateral lung was the most common site of distant metastasis (*n* = 29), followed by the brain (*n* = 10) and pleura (*n* = 8). There were 57 deaths among all patients, 34 of which were due to disease progression. As a result, the 2-year local progression-free survival (PFS), PFS, and overall survival (OS) rates for all patients were 94.0%, 71.7%, and 83.0%, respectively (Figure 1A).

In the univariate analysis, adenocarcinoma (versus others), peripheral tumor location (versus central), and BED_10_ ≥ 125 CGE were associated with improved local control. Female sex, T1 tumor (versus T2a), and BED_10_ ≥ 125 CGE were associated with better OS (Appendix A). The results of the multivariate analysis are shown in Figure 2. BED10 ≥ 125 CGE was still associated with better local control and OS. Female sex also continued to be associated with better OS. In the dose–response analysis using a generalized linear model, higher BED_10_ was significantly associated with improved local control (*p* = 0.002, Figure 3).

### 3.3. Propensity Score-Matched Analysis

To compare between survival, dosimetric parameters, and toxicities, propensity score matching was performed to create two groups of 93 patients, who each underwent photon radiotherapy or PBT. The matched population satisfied the absolute standardized mean differences of less than 10% (Appendix A). In the matched population, the local PFS (Figure 1B), PFS (Figure 1C), and OS (Figure 1D) were not significantly different. Radiation exposure to the lungs and heart was compared between groups in the matched population (Figure 4). Compared to photon radiotherapy, PBT showed significantly reduced mean radiation doses to the lung (the median value, 4.5 versus 4.1 CGE, respectively, *p* = 0.003; Figure 4A) and heart (the median value, 1.0 versus 0 CGE, respectively, *p* < 0.001; Figure 4B). Furthermore, V5 and V10 doses in the lung and heart were significantly lower in the PBT group than in the photon radiotherapy group (Figure 4C,D). Treatment-related adverse events in the matched population are shown in Table 3. No serious adverse events occurred in either group. The incidence of radiation pneumonitis and non-cardiac chest pain did not differ between groups, whereas the incidence of rib fracture was significantly lower in the PBT group than in the photon radiotherapy group (4.3% versus 16.2%, *p* = 0.014). Following treatment, the PBT group tended towards having a lower rate of lung cancer or treatment-related symptom aggravation, including non-cardiac chest pain, chest wall pain, and any respiratory symptoms comprising cough, sputum, and dyspnea, compared to the photon radiotherapy group (39.3% versus 57.0%, *p* = 0.078).

## 4. Discussion

Because of its benefits in radiation dose distribution, PBT has been considered a promising treatment modality in improving disease control and reducing treatment-related toxicities in patients with NSCLC. However, in stage I NSCLC, the superiority of PBT over photon radiotherapy, including SABR, is controversial because SABR has shown good local control and acceptable toxicities. Moreover, owing to uncertainties in the proton beam range and motion of lung cancers, there is a concern about compromised tumor control in PBT for lung cancer. In the current study, radiotherapy, including PBT and photon radiotherapy, showed encouraging local control (2-year local PFS, 94%), especially considering that a quarter of patients presented with central lung cancer. Moreover, a radiation dose higher than a BED_10_ of 125 CGE was significantly associated with better local PFS and OS in the multivariate analysis. Furthermore, in the propensity score-matched analysis, PBT reduced the mean radiation doses and low-dose bath (V_5–10_) for the lung and heart with comparable disease control and overall survival to photon radiotherapy.

In our previous dosimetric study, PBT reduced the mean radiation dose and low-dose bath (V_5–10_) for the lung and heart in all tumor locations, including the central and peripheral locations, and close to the chest wall [13]. The current study reproduced these results by comparison between delivered radiation doses in the patients treated with PBT and those in photon radiotherapy. Despite the lower radiation exposure to the lungs in PBT compared to photon radiotherapy, the incidence of radiation pneumonitis was not significantly different between groups (Table 3). In a previous prospective randomized study of stage III NSCLC, the mean lung dose and lung V_5–10_ did not differ between patients who underwent intensity-modulated radiotherapy and PBT. In our studies, PBT showed dosimetric benefits for the lung compared to photon radiotherapy, unlike the previous study on stage III NSCLC [20]. The median mean heart doses of the PBT and photon radiotherapy groups in the matched population were 1.0 and 0 CGE, respectively (*p* < 0.001). In a previous study, the rate of major coronary events increased by 7.4% with each increase in the mean heart dose of 1 Gy [21]. According to this study, utilizing PBT may significantly reduce major coronary events compared to photon radiotherapy, although long-term follow-up is needed to confirm the benefit of PBT in heart diseases. In addition to lung and heart toxicities, chest wall pain and rib fracture are relatively common adverse events in SABR for NSCLC [8,22], and the incidence of these toxicities is associated with radiation to the chest wall [23,24]. In the matched population, PBT reduced radiologic rib fractures compared to photon radiotherapy (Table 3). In the current study, analysis of radiation doses to the chest wall was unavailable; however, our previous dosimetric study showed that compared to photon radiotherapy, PBT reduced the maximum dose and delivered dose to the chest wall to 30 cc (D30cc), for tumors close to the chest wall and peripheral tumors [13]. These findings indicate that PBT reduces chest wall-related toxicity.

In the current study, a higher radiation dose of BED_10_ ≥ 125 CGE was an independent prognostic factor for better local PFS and OS in the multivariate analysis, while tumor location was not. Patients treated with a higher BED_10_ (≥125 CGE) showed better 2-year local PFS (97.2% versus 91.3%, adjusted *p*-value = 0.034) and OS (87.9% versus 74.9%, adjusted *p*-value = 0.006) rates than those treated with a moderate BED_10_ (100 CGE ≤ BED_10_ < 125 CGE; Appendix A). The survival benefits of higher doses are concordant with those of previous studies [25,26]. High-dose radiotherapy is challenging for centrally located NSCLC owing to increasing toxicities. The major complications of ablative radiotherapy for centrally located NSCLC include bronchopulmonary fistula, airway necrosis, and airway bleeding [4]. As a result, protracted dose fractionation schedules have been tested for central tumors [7]; however, ablative radiotherapy for central tumors is still challenging, especially in ultra-central locations [5]. In our previous dosimetric study, PBT showed trends for lower radiation doses to the proximal bronchial tree than photon radiotherapy, although the differences were not statistically significant [13]. Our results indicate that PBT may be useful in safe dose escalation in central tumors.

In the current retrospective multi-institutional study, although we used multivariable Cox proportional hazards models and propensity score-matched analysis to compare PBT and photon radiotherapy, there may be confounding bias. Furthermore, grade 3 or higher treatment-related toxicities were only observed in seven patients who presented with grade 3 radiation pneumonitis; however, we generally used compromised dose-fractionated schedules with a lower BED_10_ of less than 100 CGE for central tumors to avoid complications. Owing to the retrospective nature of the current study, there is potential for underestimation of treatment-related toxicities. Despite these limitations, to our knowledge, the current study is the largest to compare PBT with photon radiotherapy for stage I NSCLC. Furthermore, the dosimetric benefits of PBT shown here are concordant with those reported in our previous dosimetric study. Finally, PBT did not worsen disease control despite its various potential uncertainties.

## 5. Conclusions

In conclusion, we demonstrated that PBT significantly reduced radiation exposure to the heart and lungs without worsening disease control. The outcomes after PBT and photon radiotherapy with a higher BED for stage I NSCLC were satisfactory. We will continually strive to improve the outcomes for central tumors by defining optimal dose fractionation schedules using PBT.

## Figures and Tables

**Figure 1 cancers-14-03627-f001:**
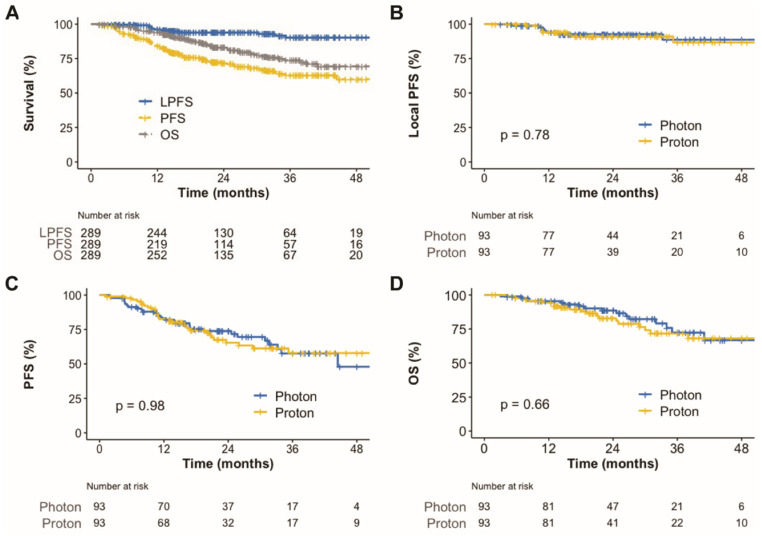
Kaplan–Meier plot of local progression-free survival, progression-free survival, and overall survival rates for all patients over 48 months (**A**). Kaplan–Meier plot of local progression-free survival (**B**), progression-free survival (**C**), and overall survival rates (**D**) for matched patients according to radiotherapy techniques over 48 months. PFS, progression-free survival; OS, overall survival.

**Figure 2 cancers-14-03627-f002:**
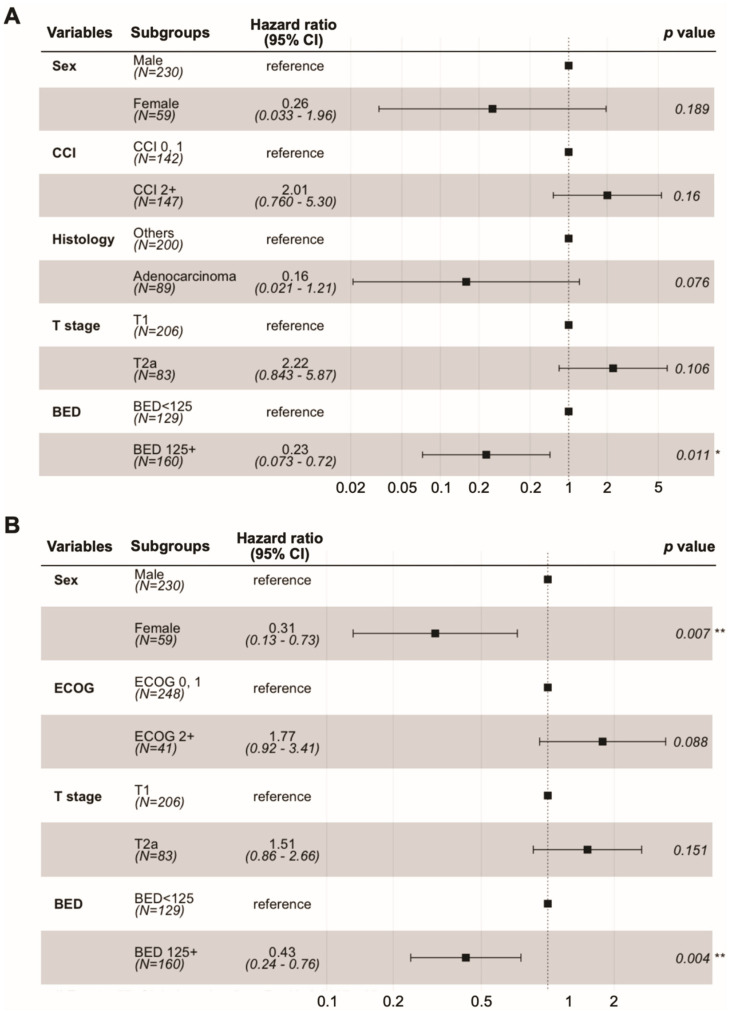
Forest plots of Cox proportional hazards regression model. Local progression-free survival (**A**). Overall survival (**B**). * <0.05; ** <0.01.

**Figure 3 cancers-14-03627-f003:**
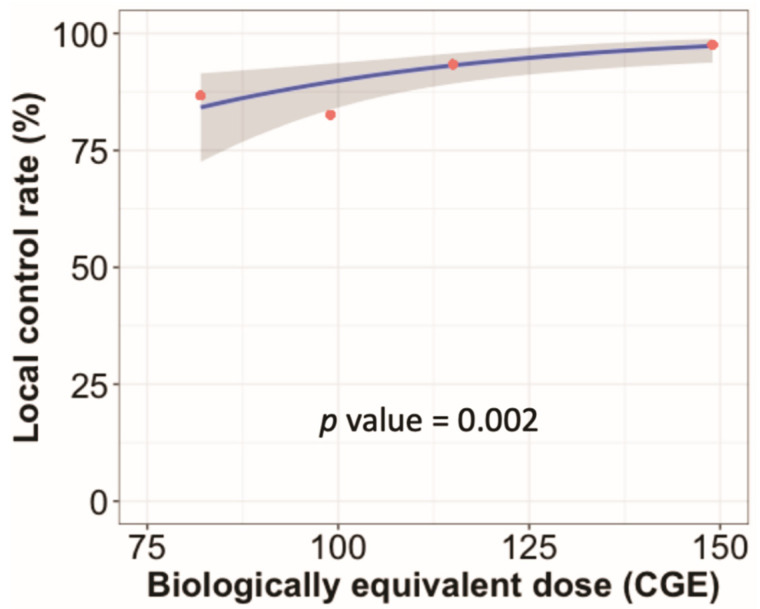
Radiation dose–response analysis. The blue line indicates the predicted local control rate determined by a generalized linear model according to the biologically equivalent dose using α/β = 10 Gy (BED_10_). The red circles indicate estimated local control rates for groups by radiation dose categories. The radiation dose categories were BED_10_ of 75–90 (*n* = 45), 90–110 (*n* = 23), 110–130 (*n* = 61), and 130–150 CGE (*n* = 160). The estimated local control rates for these categories were 86.7%, 82.6%, 93.4%, and 97.5%, respectively.

**Figure 4 cancers-14-03627-f004:**
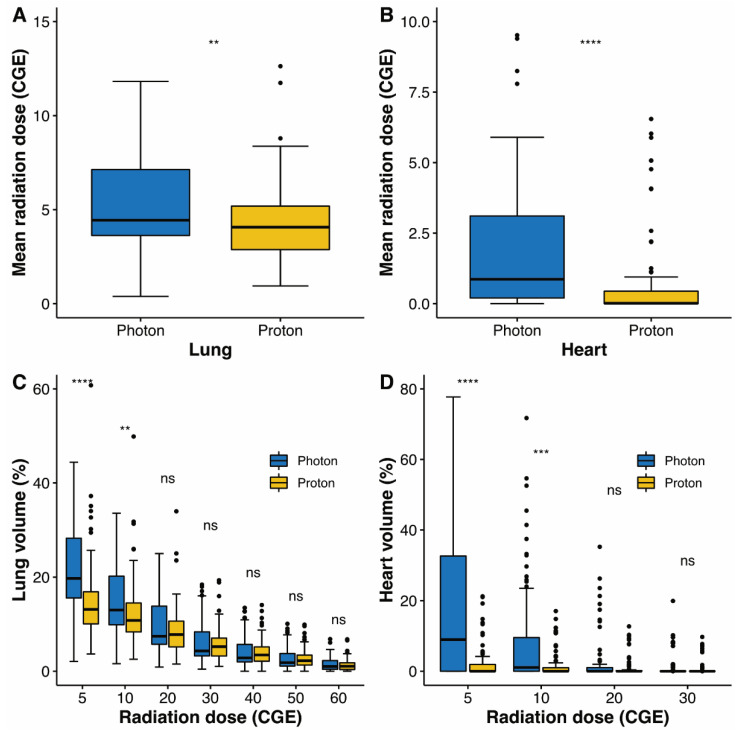
Box plot of mean radiation doses to the lung (**A**) and heart (**B**). Box plot of distributions of dose–volume indices for the lung (**C**) and heart (**D**). Whiskers indicate 1.5 times the interquartile range above and below the mean; dots represent individual observation. CGE, cobalt gray equivalent; ns, non-significant; ** <0.01; *** <0.001; **** <0.0001.

**Table 1 cancers-14-03627-t001:** Patients’ characteristics.

Characteristics	All, No. (%) (*n* = 289)	Photon, No. (%) (*n* = 177)	Proton, No. (%) (*n* = 112)	*p*-Value
Age, years *	76 (72–80)	77 (72–81)	75 (70–80)	0.235
Sex				
Male	230 (79.6)	139 (78.5)	91 (81.2)	0.683
Female	59 (20.4)	38 (21.5)	21 (18.8)	
Smoking history				
Never	65 (22.5)	42 (23.7)	23 (20.5)	0.052
Former	182 (63.0)	103 (58.2)	79 (70.5)	
Current	42 (14.5)	32 (18.1)	10 (8.9)	
ECOG performance status				
0	35 (12.1)	21 (11.9)	14 (12.5)	0.004
1	213 (73.7)	121 (68.4)	92 (82.1)	
2	37 (12.8)	31 (17.5)	6 (5.4)	
3	4 (1.4)	4 (2.3)	0	
Charlson comorbidity index				
0	56 (19.4)	32 (18.1)	24 (21.4)	0.494
1	86 (29.8)	58 (32.8)	28 (25.0)	
2	63 (21.8)	39 (22.0)	24 (21.4)	
3+	84 (29.1)	48 (27.1)	36 (32.1)	
Chronic lung disease				0.119
No	113 (39.1)	76 (42.9)	37 (33.0)	
Yes	176 (60.9)	101 (57.1)	75 (67.0)	
Baseline FEV1, % predicted *	78.0 (59.0–96.0)	78.5 (62.0–97.0)	76 (55.0–92.0)	0.324
Baseline DLCO, % predicted ^†^	65.0 (21.2)	68.3 (20.7)	64.3 (21.9)	0.137

ECOG, Eastern Cooperative Oncology Group; FEV1, forced expiratory volume in 1 s; DLCO, diffusion capacity for carbon monoxide. * Data are median (interquartile range). ^†^ Data are mean (standard deviation).

**Table 2 cancers-14-03627-t002:** Tumor and treatment characteristics.

Characteristics	All, No. (%) (*n* = 289)	Photon, No. (%) (*n* = 177)	Proton, No. (%) (*n* = 112)	*p*-Value
Tumor histologic type				
Squamous cell carcinoma	89 (30.8)	53 (29.9)	36 (32.1)	0.263
Adenocarcinoma	88 (30.5)	48 (27.1)	40 (35.7)	
Others	11 (3.8)	7 (4.0)	4 (3.6)	
Unproven	101 (34.9)	69 (39.0)	32 (28.6)	
Tumor location				
Peripheral	56 (19.4)	36 (20.3)	20 (17.9)	0.454
Close to chest wall	157 (54.3)	99 (55.9)	58 (51.8)	
Central	76 (26.3)	42 (23.7)	34 (30.4)	
Tumor lobar location				0.033
Left upper lobe	67 (23.2)	44 (24.9)	23 (20.5)	
Left lower lobe	74 (25.6)	36 (20.3)	38 (33.9)	
Right upper lobe	87 (30.1)	61 (34.5)	26 (23.2)	
Right middle lobe	14 (4.8)	6 (3.4)	8 (7.1)	
Right lower lobe	47 (16.3)	30 (16.9%)	17 (15.2)	
T stage *				
T1a	21 (7.3)	11 (6.2)	10 (8.9)	0.802
T1b	94 (32.5)	60 (33.9)	34 (30.4)	
T1c	91 (31.5)	56 (31.6)	35 (31.2)	
T2a	83 (28.7)	50 (28.2)	33 (29.5)	
Total radiation dose, BED_10_ ^†^ (CGE)				
75–100	60 (20.8)	36 (20.3)	24 (21.4)	<0.001
100–125	69 (23.9)	24 (13.6)	45 (40.2)	
125–150	160 (55.4)	117 (66.1)	43 (38.4)	
Volume of ITV (cm^3^) ^‡^	29.3 (14.5–54.3)	29.9 (15.6–54.4)	28.4 (12.9–52.9)	0.612

BED, biologically equivalent dose; CGE, cobalt gray equivalent; ITV, internal target volume. * American Joint Committee on Cancer Staging, 8th edition; ^†^ biologically equivalent dose using an α/β ratio of 10 Gy; ^‡^ data are median (interquartile range).

**Table 3 cancers-14-03627-t003:** Treatment-related adverse events.

	Photon, No. (%) (*n* = 93)	Proton, No. (%) (*n* = 93)	*p*-Value
Radiation pneumonitis			
Grade 0	9 (9.7)	12 (12.9)	0.280
Grade 1	73 (78.5)	74 (79.6)	
Grade 2	10 (10.8)	4 (4.3)	
Grade 3	1 (1.1)	3 (3.2)	
Non-cardiac chest pain and chest wall pain			
Grade 0	70 (75.3)	79 (84.9)	0.240
Grade 1	20 (21.5)	12 (12.9)	
Grade 2	3 (3.2)	2 (2.2)	
Rib fracture			
Grade 0	78 (83.9%)	89 (95.7%)	0.014
Grade 1	14 (15.1%)	4 (4.3)	
Grade 2	1 (1.1)	0 (0.0)	
Aggravation of symptoms after treatment *	53 (57.0)	40 (39.3)	0.078

* Non-cardiac chest pain, chest wall pain, and any respiratory symptoms including cough and dyspnea.

## Data Availability

The data for this study, though not available in a public repository, will be made available to other researchers upon reasonable request to the corresponding author.

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
