# Peer review of "Proton Beam Therapy versus Photon Radiotherapy for Stage I Non-Small Cell Lung Cancer"

_cancers, 2022, doi:10.3390/cancers14153627_

Round 1
Reviewer 1 Report
I thank the editors for the opportunity to review the article entitled "Proton Beam Therapy Versus Photon Radiotherapy for Stage I Non-Small Cell Lung Cancer."
In the manuscript, the authors made a comparison between proton therapy and photon radiotherapy through analysis of data from patients treated at two institutions.
The analysis showed an advantage of proton therapy regarding dose reduction to organs at risk (particularly lungs and heart).
The article would be worthy of publication, but some points need to be better explained.
-Please better explain concepts from lines 152 to 158.
-Please expound more clearly paragraphs about survival (particularly lines 170 to 172) and propensity score-matched analysis (particularly lines 210 to 212).
Author Response
-Please better explain concepts from lines 152 to 158.
Response: We changed those sentences as you indicated.
-Please expound more clearly paragraphs about survival (particularly lines 170 to 172) and propensity score-matched analysis (particularly lines 210 to 212).
Response: We added descriptions about survival and corrected errors in patient numbers. We also changed description in the “Propensity score-matched analysis” section as you indicated.
Reviewer 2 Report
This issue is important, but there are some points to revise before publishing.
1. The dose prescription data about photon radiotherapy and PBT. Could you make new table about it?
2. The dose per fraction for central lung cancer was smaller than peripheral lung cancer, as you showed. I think the more differences dose per fraction between photon and PBT, the larger the error of LQ model was. So, could you show only data of peripheral lung cancer, if possible?
3. The dose per fraction was very variable. So, the mean of value of dosimetric parameter is less than other clinical data. What do you think?
Author Response
- The dose prescription data about photon radiotherapy and PBT. Could you make new table about it?
Response: We made a new table (Supplementary Table 1) for the dose prescription data according to photon radiotherapy and PBT as well as tumor location.
- The dose per fraction for central lung cancer was smaller than peripheral lung cancer, as you showed. I think the more differences dose per fraction between photon and PBT, the larger the error of LQ model was. So, could you show only data of peripheral lung cancer, if possible?
Response: Because radiation dose prescription and prognosis are different between central and peripheral tumors, differences between photon radiotherapy and PBT could be masked as your comment. However, to eliminate this bias, we performed multivariate Cox proportional hazards analyses (Figure 2 and Supplementary Table 2) and propensity score-matched analysis. In the multivariate analysis, tumor location (central vs. peripheral) was not a significant predictor for local progression-free survival and overall survival (tumor location was rejected from the final Cox proportional hazard regression model due to its insignificance, so it was not shown in Figure 2 and Supplementary Table 2.). In the propensity score-matched analysis, the proportion of central and peripheral tumors was well-balanced between photon radiotherapy and PBT groups. Therefore, tumor location may not impact the results of our study. So, we hope we can maintain our study population.
- The dose per fraction was very variable. So, the mean of value of dosimetric parameter is less than other clinical data. What do you think?
Response: Although dose per fraction was variable between patients, total doses ranged from 48 to 70 Gy, and almost patients (86.4%) received 60 Gy or more. Therefore, we think various radiation doses did not influence smaller mean values of dosimetric parameters. In our study, more than half of the patients had the tumor close to the chest wall. This tumor location characteristic may be the reason for smaller the mean values of dosimetric parameters than in other studies.
Reviewer 3 Report
Your study was conducted using appropriate methods and the results were convincing. Other differences might be observed in the future with longer follow-up periods and quality of life assessments.
Lines 26-27.
The English wording may be a little confusing.
Please confirm.
(For example, "where the only two facilities in South Korea that PBT is available.”)
Author Response
Lines 26-27.
The English wording may be a little confusing.
Please confirm.
(For example, "where the only two facilities in South Korea that PBT is available.”)
Where PBT is available in only two facilities in South Korea
Response: We changed that sentence as your suggestion.